# Mechanical Stability of a Small, Highly-Luminescent Engineered Protein NanoLuc

**DOI:** 10.3390/ijms22010055

**Published:** 2020-12-23

**Authors:** Yue Ding, Dimitra Apostolidou, Piotr Marszalek

**Affiliations:** 1Department of Mechanical Engineering and Materials Science, Duke University, Durham, NC 27708, USA; yueding@xjtu.edu.cn (Y.D.); da150@duke.edu (D.A.); 2Department of Engineering Mechanics, SVL, Xi’an Jiaotong University, Xi’an 710049, China

**Keywords:** NanoLuc, Single Molecule Force Spectroscopy, bioluminescence, protein folding, AFM, protein mechanics

## Abstract

NanoLuc is a bioluminescent protein recently engineered for applications in molecular imaging and cellular reporter assays. Compared to other bioluminescent proteins used for these applications, like Firefly Luciferase and Renilla Luciferase, it is ~150 times brighter, more thermally stable, and smaller. Yet, no information is known with regards to its mechanical properties, which could introduce a new set of applications for this unique protein, such as a novel biomaterial or as a substrate for protein activity/refolding assays. Here, we generated a synthetic NanoLuc derivative protein that consists of three connected NanoLuc proteins flanked by two human titin I91 domains on each side and present our mechanical studies at the single molecule level by performing Single Molecule Force Spectroscopy (SMFS) measurements. Our results show each NanoLuc repeat in the derivative behaves as a single domain protein, with a single unfolding event occurring on average when approximately 72 pN is applied to the protein. Additionally, we performed cyclic measurements, where the forces applied to a single protein were cyclically raised then lowered to allow the protein the opportunity to refold: we observed the protein was able to refold to its correct structure after mechanical denaturation only 16.9% of the time, while another 26.9% of the time there was evidence of protein misfolding to a potentially non-functional conformation. These results show that NanoLuc is a mechanically moderately weak protein that is unable to robustly refold itself correctly when stretch-denatured, which makes it an attractive model for future protein folding and misfolding studies.

## 1. Introduction

How proteins fold to their unique three-dimensional structures and how they maintain the structural conformations that are required for their biological function are central questions in biophysics [1,2,3,4,5,6,7,8]. These processes underlie many important phenomena such as those related to cellular homeostasis [9,10,11,12,13], cancer metabolism [1,14,15,16,17], and the onset and treatment of protein misfolding diseases [1,18,19,20]. Single-molecule force spectroscopy (SMFS) techniques are particularly powerful tools for studying protein folding because they provide a means to directly apply forces to individual proteins under native conditions in order to measure their structural response and the internal forces that stabilize the protein [21,22,23,24,25,26,27,28,29,30,31,32,33,34,35,36,37,38,39,40,41,42,43,44,45,46,47,48,49,50,51,52,53,54,55,56,57,58,59,60,61,62,63,64]. In these experiments, numerous “stretch” and “relax” cycles of force application are performed on the same protein to examine molecular elasticity within different extensions, tension regimes, and loading rates. At sufficient large extensions, the resulting tension may drive structural rearrangements within the molecule, termed force-induced conformational transition. These rearrangements that lead to high-energy conformations may reveal extremely interesting molecular properties that are not accessible to typical spectroscopic methods that usually examine biomolecules at or near their equilibrium states. As the forces are relaxed, the protein might also “re-fold”, providing insights into the conditions or requirements for these recovery events to occur.

Understanding how these applied and destabilizing forces might affect not only the structure but catalytic activity of those proteins can provide new insights into the molecular mechanisms of how misfolding, aggregation, or recovery to their functional forms with the help of other chaperone proteins can contribute to cellular health or disease [65,66,67,68,69,70,71,72,73]. However, SMFS with simultaneous measurement of catalytic activity of the protein to which forces are being applied has been rather challenging due to numerous technical difficulties [44,74,75,76,77].

Here, we report on the mechanical unfolding and refolding properties of a promising candidate for combined mechanical and catalytic measurements at a single-molecule level, a bioluminescent protein NanoLuc. NanoLuc is a 19.1 kDa enzyme that catalyzes the conversion of furimazine to furimamide, a process that emits visible light that can be detected [78]. While there have been a few SMFS studies on bioluminescent proteins including Firefly Luciferase (FLuc) in the past [53,67,79], FLuc bioluminescence is likely too weak to be registered at a single molecule level using current detection technologies; NanoLuc is able to emit 150× more light compared to FLuc [78,80]. We expect that unraveling of the tertiary structure of NanoLuc due to mechanical unfolding should be accompanied by a significant decrease in its bioluminescence activity. Conversely, we expect that successful NanoLuc refolding should restore its ability to produce bright light similar to what is observed in bulk experiments for FLuc [81,82]. Thus, in addition to the nanomechanical unfolding and refolding properties of NanoLuc reported for the first time, this work represents a step toward a new assay for combined nanomechanical and catalytic/luminescence single-molecule studies and as a potential new and powerful substrate system for protein misfolding and chaperone research.

## 2. Results and Discussion

### 2.1. NanoLuc Protein for Force Spectroscopy

For SMFS measurements, the protein of interest needs to have molecular handles attached to apply the mechanical force and separate the protein from the surface and force probe minimizing the direct interaction with the instrument that could be potentially denaturing [83]. In addition, the mechanical properties of the handles produce an unmistakable “force spectrogram” fingerprint allowing one to differentiate truly single molecule recordings from recordings obtained on multimolecular assemblies [84]. Frequently, instead of using a monomeric protein of interest, tandem repeats of the same protein (polyproteins) are constructed in order to produce a repetitive and consistent pattern in the force extension data that is easy to identify. Polyproteins are particularly useful for low force unfolding events, which could be masked by nonspecific adhesive interactions at the beginning of the force–extension curve or confused with instrumental drift and noise. An additional benefit of using polyproteins is the improved efficiency of data collection in proportion to the number of repeats of the protein of interest. This is significant considering that SMFS is a low-throughput technique [83,84] because the likelihood of picking up a single molecule for mechanical manipulations is typically lower than 1%. For those reasons, we generated a “polyprotein” gene for three tandem repeats of NanoLuc and flanked them at the DNA level with two I91 (formerly I27) domains at each end, which serve as SMFS handles and reference proteins allowing to identify truly single-molecule recordings based on the characteristic mechanical unfolding fingerprint of poly(I91) proteins [64,85]. When expressed, force spectroscopy on the polyprotein allowed, for every successful observation of the onset of protein unfolding at the “single molecule” level, three-fold increase in the number of mechanical denaturing events that could be analyzed. We expressed I91_2_-NanoLuc_3_-I91_2_ construct in *E. coli*. After purification, the chimera protein containing three NanoLuc repeats was actively bioluminescent (Figure 1a). At this point, we have no comparison of bioluminescence intensity of this construct with a monomeric isolated NanoLuc. A detailed comparison of catalytic activity of polyNanoLuc versus monomeric NanoLuc is warranted and it will require constructing a polyNanoLuc protein without I91 handles.

In order to quantify the bioluminescence of our construct, we performed various dilutions starting at 10 nM (top right) and ending at ~16 fM (bottom left) and observed a linear behavior for our data at room temperature (Figure 1c). Since NanoLuc is 150x brighter than FLuc, these results demonstrate that our construct can be used for single molecule studies of catalysis for concentrations as low as 16 fM. Furthermore, the linear trend indicates that for these concentrations we are in the linear dynamic range of our bioluminometer with a proportional relationship between luminescence and enzyme concentration [86]. Above 10 nM high light intensities would saturate the bioluminometer.

Additionally, we examined the time decay of the bioluminescence of 10 nM I91_2_-NanoLuc_3_-I91_2_ over time by collecting measurements at different time points at room temperature (Figure 1d). We observe an exponential decay, caused by substrate consumption [86]. We calculated that I91_2_-NanoLuc_3_-I91_2_ construct has a decay of *t_1/2_* = *58 min*.

### 2.2. Mechanical Unfolding of NanoLuc

In the atomic force microscope (AFM)-based SMFS that we used in this study, one end of the polyprotein is attached to a surface (through nonspecific adsorption) while the other end is attached to the tip of an AFM micro-cantilever, which is used as a force sensor. In the case of AFM-based SMFS the protein is extended by a piezoelectric actuator supporting the sample, with the precision better than 1 nm (10^−9^ m), while the applied force (tension) is measured by accurately determining the deflection of the end of the cantilever, using a laser beam, reflected off the top of the cantilever and projected onto a position-sensitive photodiode (see Materials and Methods). The AFM force precision ranges from around 10 pN (10^−12^ N) for most commercial instruments and cantilevers, and it may approach 1 pN for specially micromachined custom cantilevers [87]. By relating the deflection of the AFM cantilever to displacement of the surface, an applied force vs. protein extension curve is generated that reveals how the protein deforms in response to force, and how segments of the protein might denature, which is observed by a sudden drop (a “peak”) in the force–extension curve. The pathway by which the NanoLuc polyprotein unfolds was determined by applying load with a range of constant velocities between 0.05–0.25 nm/ms (most recordings were at 0.25 nm/ms). When performing force spectroscopy on the I91_2_-NanoLuc_3_-I91_2_ construct, we observe up to four peaks or unfolding events that are characteristic of the four I91 domains denaturing in response to applied forces in our recordings (cyan) along with three other distinct smaller peaks (red), which suggest one unfolding event per NanoLuc domain (Figure 2). To identify the number of amino acids of the polyprotein involved in each denaturation event from the “contour length” of the denatured segment [27], prior to our quantitative analysis we determined the persistence length of NanoLuc by adopting the worm-like chain (WLC) model to fit the force–extension curve and using the root mean square error approach to determine a persistence length of 0.4 nm (results not shown). We later used persistence length of 0.4 nm in our analysis for NanoLuc force–extension curves.

As the AFM cantilever is brought into contact with a surface coated with the polyprotein and retracted, several typical unfolding traces of I91_2_-NanoLuc_3_-I91_2_ construct are shown in Figure 3a–d. The complete unfolding force–extension curve of our entire construct is shown in Figure 3a, which includes three smaller peaks for the NanoLuc proteins preceding four large peaks for the I91 domains. However, because the proteins adsorb to the AFM tip randomly along their length prior to extension, other unfolding peak combinations are possible with the number of I91 peaks less than four. In principle, only the recordings that captured at least three I91 unfolding peaks guarantee that all three NanoLuc proteins were also subjected to the applied force (this is evident from the design of the I91_2_-NanoLuc_3_-I91_2_ construct). This strong criterion allows to establish the mechanical unfolding fingerprint of NanoLuc_3_. Having established such a “template” recording it is then possible to relax the criterion involving three I91 peaks and we could also include in the data analysis force–extension curves that captured less than three I91 peaks (e.g., 1 or 2) as long as the preceding smaller peaks overlap reasonably well with the NanoLuc_3_ template that is characterized by peak force magnitudes and peak spacing. To illustrate this approach, we overlaid the recordings with less than three I91 peaks (Figure 3b,d) onto the template recordings with four I91 peaks in Figure 3e. All these force curves are well fitted by the WLC model (shown in dashed lines) with the comparable contour length increment strongly suggesting that they were recorded on single protein molecules harboring three NanoLuc repeats.

By the changes in the fit parameters for the length of the polymer (contour length increments) subjected to applied forces before and after an ‘unfolding’ event, we can estimate the number of amino acids that were contributing to the denaturation by the applied force. The contour length increments from WLC model fitting and the unfolding forces are shown in a scatter plot (Figure 4a) for both NanoLuc protein and I91 domains. We selected 54 recordings with at least three I91 domains unfolding events in each recording, ensuring that all three NanoLuc proteins were fully unfolded. From the scatter plot we observe two populations that are clustered, one for I91 domains (pink) and one for NanoLuc (blue). Normalized histograms of contour length increment (nm) and unfolding force (pN) of NanoLuc are presented in Figure 4b,c, respectively. Bin sizes were determined using Sturge’s formula, $k=\lceil log_{2}n\rceil +1$, where k is the number of bins and n is the sample size [88]. Fitting these results with normal distribution gives a contour length increment of 64.48 ± 0.64 nm (mean ± SE, *n* = 88) and an unfolding force of 71.84 ± 2.86 pN (mean ± SE, *n* = 88), which makes NanoLuc a moderately mechanically weak protein. For comparison, ankyrin repeat proteins unfold at around 20–25 pN [89] and spectrin domains unfold at around 25–35 pN at comparable stretching speeds (~0.3 μm/s) [90]. Both are examples of mechanically weak proteins. On the other hand, some proteins, such as titin domains from muscle, unfold at around 200 pN at the same stretching speeds and those were considered as mechanically strong proteins until the Gaub group recently identified bacterial adhesion proteins that unfold at amazingly high forces of 2000 pN [91]. We note that the distribution in Figure 4c displays non-Gaussian features, so further measurements are warranted to clarify their origin. We theoretically calculated the contour length increment of a protein by using the total length of the residues subtracts the initial length between the N terminus and C terminus [27]. For NanoLuc, its contour length increment is 169 residues × 0.365 nm/residue − 0.262 nm = 61.42 nm, which is consistent with our experimental results. Similarly, I91 domains give 28.07 ± 0.25 nm (mean ± SE, *n* = 195) for contour length increments and 227.88 ± 3.30 pN (mean ± SE, *n* = 195) for rupture forces, which are consistent with the previous results for the unfolding of the I91 (histograms not shown) [92]. 

### 2.3. Cyclic Measurements of NanoLuc Refolding and Misfolding

In a “cyclic” SMFS measurement, the forces applied to a protein are briefly relaxed to allow the protein the opportunity to refold prior to subsequent application of mechanical forces: if they are able to successfully refold during that time, the force–extension curves will appear similar as those before the first mechanical denaturing, otherwise the force–extension behavior will appear like that of an unstructured polypeptide or exhibit unusual/unpredictable behavior indicating that the protein misfolded to an improper conformation during that time. The cyclic measurements are conducted at a constant stretching rate of ~0.05 nm/ms. After a single molecule was initially denatured by pre-stretching to an extension of 200–280 nm, after which all three NanoLucs initially denatured, the AFM tip was returned to a position close to the surface without touching and then stretched the molecule again with a smaller extension (150–180 nm). The smaller extension is chosen to avoid I91 unfolding and a possible interaction between NanoLuc and I91 during refolding attempts. Each refolding attempt spans 6–8 s and, during the process, the force remains under 5 pN for ~1.6 s to allow the polyprotein to refold [53]. Figure 5a shows six representative cycles of cyclic measurements on the same molecule. The complete unfolding curve is plotted (blue lines) as the template to analyze the refolding attempts of each cycle. In cycle (1), two NanoLuc peaks overlap well with the template during the stretch, which indicates the NanoLuc proteins are correctly refolded. There is only one peak appearing in cycle (32) located after the third peak of NanoLuc in the template, which is considered to be a misfolding event. In the following cycle (33), no peak shows during the measurement meaning no refolding or misfolding event happens in this cycle. One NanoLuc correctly refolds in cycle (38), which can be confirmed by the peak showing at the third peak of NanoLuc in the template. Both cycle (43) and (44) show misfolding events with either a peak located close to the second peak (cycle (43)) or between the second and the third peaks (cycle (44)) of the template.

We performed 635 refolding attempts on 21 single molecules of I91_2_-NanoLuc_3_-I91_2_ and there are 248 attempts showing clear peaks in the force–extension curves. The distribution of the contour length increment for the refolding experiments (*n* = 248) is shown in the normalized histogram (Figure 5b). Comparing with unfolding results (blue, *n* = 88), the contour length increment from refolding experiments is more widely distributed due to the occurrence of misfolded proteins during refolding attempts. A normal distribution is adopted to fit the contour length increment of refolding tests and the contour length increment is 60.09 ± 1.67 nm (mean ± SE, *n* = 268).

All of the 635 refold recordings can be grouped into three categories: no peaks (56.2%, *n* = 357), correctly refolded events (16.9%, *n* = 107) and misfolded events (26.9%, *n* = 171). The correctly refolded group contains curves with either one peak or two peaks from refolded proteins. The misfolded cases are classified into three different groups: the curve has only one peak with a contour length increment smaller than NanoLuc (12.3%, *n* = 78), the curve has a peak with a contour length increment about 1.5 times of that of NanoLuc (5.0%, *n* = 32) and the curve has other types of misfolded events (9.6%, *n* = 61). 

In Figure 6a, we show 10 force–extension curves that have one peak during each refolding attempt from the correctly refolded proteins. All of them are superimposed at the bottom and the peaks overlap well with the unfolding event of NanoLuc in the template (black curve). The histogram of contour length increment for the group of correctly refolded proteins is shown in Figure 6c. The distribution can be fit by a normal distribution with 59.61 ± 1.18 nm (mean ± SE, *n* = 117), which is consistent with the contour length increment of the unfolding events.

Figure 6b shows 10 examples of refolding attempts of I91_2_-NanoLuc_3_-I91_2_ from the group with the first type of misfolded event. Each curve has a peak showing a smaller contour length increment than NanoLuc and matches well with each other when superimposed together at the bottom. The difference between these types of peaks and the normal NanoLuc peaks can be observed at the bottom. This group dominates the left part of the distribution of contour length increment from distribution of the whole refolding attempt (in gray), as shown in Figure 6d, and has a contour length increment of 36.85 ± 1.05 nm (mean ± SE, *n* = 78).

The second type of misfolding event can be seen in Figure 7a. In this group, there are 32 recordings and 10 of them are plotted in this figure. The peak in the superimposed curve located between the second and third peaks of NanoLuc in the template. The distribution is shown in Figure 7c with the contour length increment at 91.31 ± 1.68 nm (mean ± SE, *n* = 32), which is 1.5 times that of NanoLuc. This type of misfolding event may be caused by the interaction between two NanoLuc proteins.

Six curves with other types of misfolding behavior are shown in Figure 7b. The peak in the first recording lies between the first peak and second peak in the template. As a result, the contour length increment of this peak is two–three times the contour length increment of NanoLuc. Similarly, the second recording has a contour length increment twice that of NanoLuc. As can be seen in the third, fourth and fifth curves, there might be two misfolded events appearing in each refolding attempt and this may lead to smaller contour length increments. We also found some curves have many nonspecific peaks during refolding attempts (like the last one in Figure 7b) and these recordings are not included in the analysis. Due to the different types of misfolding behavior, the distribution of contour length increment (82.42 ± 7.02 nm, mean ± SE, *n* = 40) of this group (Figure 7d) is more distributed than other groups. Overall, it appeared that NanoLuc was unable to refold robustly on its own and exhibited a variety of putatively misfolded structures during the cyclic measurements.

## 3. Materials and Methods 

### 3.1. Protein Purification

NanoLuc sequence was acquired from Integrated DNA Technologies Inc. (Research Triangle Park, NC, USA), based on PDB file 5IBO. The gene sequence was cloned into plasmid pEMI91 by GenScript Biotech (Piscataway, NJ, USA), resulting in a final construct of I91_2_-NanoLuc_3_-I91_2_ [93]. All four I91 domains and three NanoLuc domains had shuffled codons. The entire construct has 930 amino acids, 89 amino acids per I91 domain and 170 amino acids per NanoLuc protein with the rest being linkers and purification tags. The full nucleic acid (5′ to 3′) and amino acid sequence (N to C-terminus) of I91_2_-NanoLuc_3_-I91_2_ construct is shown using the DNASTAR program (Version 17.1, Madison, WI, USA) in Figure 8. Brown bars are used to indicate the different proteins in the construct, while a light yellow bar is used for the amino acid sequence. Purification tags, such as HisTag and StrepTag, are marked with a blue arrow.

Protein was expressed using BL21-Gold (DE3) Competent cells (Agilent, Santa Clara, CA, USA) grown at 37 °C until 0.8 OD, followed by induction at room temperature with 1 mM IPTG for 3 h. The cells were then centrifuged, and their pellet was flash frozen, prior to chemical lysis. The lysis buffer consisted of binding buffer (50 mM Phosphate Buffer, 300 mM NaCl, 20 mM imidazole at pH 8.0), protease inhibitor cocktail EDTA-Free, 1 mg/mL lysozyme, 3 units/mL benzonase, 1 mM DTT, 0.5% Triton X-100. We used 50 mL lysis buffer per 1 L culture. Protein was collected by binding to a HisTag column (Ni-NTA Agarose-Qiagen, Germantown, MD, USA). The purified protein was flash frozen at concentrations of 1–2 mg/mL in 25 mM Hepes, 100 mM KCl, 1 mM DTT (Buffer A).

### 3.2. Atomic Force Microscopy

Force spectroscopy experiments were performed consistently with the protocol previously described [92]. A custom-built Atomic Force Microscope (AFM) was used for all experiments (Figure 9). AFM cantilevers OBL and MCLT (Bruker Corporation, Camarillo, CA, USA) were used for the force spectroscopy experiments, with spring constants ~6 and ~23 pN/nm, respectively. The spring constant of each cantilever was determined using the energy equipartition theorem [94].

For our measurements, the NanoLuc construct was diluted to 0.25 mg/mL in Buffer A, and 50 μL of the protein solution was loaded on a gold coated glass slide (Ted pella, Redding, CA, USA) for 30–40 min. The unbound proteins were removed by pipetting in and out 10 μL of Buffer A 3 times so the surface remained solvated. The cantilever of the AFM was held at a distance (350–500 nm) from the protein covered surface with the sample (separated) and brought in contact. During contact, if a single molecule was attached to the tip of the cantilever, while separating the cantilever from the surface, the deflection of the cantilever reflected the amount of force applied to the protein to which it was bound (force pulling on the protein) as the cantilever was retracted away from the surface at a constant velocity 0.05–0.25 nm/ms.

Cyclic measurements, which include repetitive unfolding and refolding of the protein molecule, were performed. These experiments were also performed like described above with two differences: the cantilever was separated from the surface for a shorter distance 150–280 nm (only part of the complete construct was unfolded) and the cantilever was brought close to the surface without touching (5–10 nm away from surface). This prevented multiple molecules from being attached to the cantilever, which would result in interference of the unfolding signature by the parallel unfolding of these molecules. In all cyclic measurements the initial unfolding signature included one or two NanoLuc peaks, overlapping correctly with the full construct. We avoided unfolding I91 domains to avoid interference during refolding of NanoLuc proteins. Therefore, when the cantilever was very close to the surface, the proteins were able to refold due to the stretching forces applied to the molecule being below 5 pN. Also, by no contact to the surface, we prevented the attachment of multiple molecules to the cantilever. This is an important step so that multiple molecules do not unfold simultaneously, giving confusing recordings that are very hard to interpret. Thus, we were able to study the refolding of the proteins in our construct, when they were left to relax for ~1.6 s. All unfolding curves were thus from a single molecule, which was repetitively relaxed and pulled. This was repeated multiple times on the same protein molecule, for various molecules at a time.

In the beginning of the cyclic measurements (1st round of cyclic), we separated the tip from the surface at a distance of 200–280 nm based on the amino acid sequence of the different proteins in the construct [27], so that three NanoLuc with perhaps one I91 domain unfolded. This distance was empirically found to give the best results in successfully picking single molecules to proceed with the next rounds of cyclic measurements.

All force–extension curves were analyzed using MatLab R2019a (Mathworks, Natick, MA, USA). The worm-like chain (WLC) model was used to fit the force extension curves through LabVIEW platform (National Instruments Corporation, Austin, TX, USA) for the contour length of the protein subjected to forces, the increase in the contour length after a sudden drop in forces resulting from a protein domain unfolding, and the forces at which these unfolding events occurred. The interpolation formula for the WLC model is $\frac{Fp}{k_{B}T}=\frac{1}{4}{\left(1-\frac{x}{L}\right)}^{-2}-\frac{1}{4}+\frac{x}{L}$, where F is the Force, p is the persistence length, k_B_T is 4.114 pNnm, x is the extension, and L is the contour length [95]. To determine the persistence length, we fitted this equation to our recordings of I91_2_-NanoLuc_3_-I91_2_ construct with three NanoLuc peaks and four I91 peaks, indicative of measurements where molecules are stretched by the termini. Different values of persistence length in the range of 0.2–0.6 nm were tried and we found that NanoLuc peaks were best fitted with persistence length of 0.4 nm, which produced the smallest standard deviation of contour length increment $\Delta L_{c}$. The choice of force–extension curves for further analysis for the mechanical characteristics of NanoLuc was based on the presence of the force–extension signal and protein domain unfolding events typical of the four titin I91 domains, which with the characteristic persistence length of ~0.3 nm, contour length increment of ~28 nm, and unfolding forces of ~200 pN [96], are very well characterized. When a force–extension curve contained these characteristics, we proceeded to the next step of our analysis, including identifying the contour length and unfolding force values for other features in the curves that were assigned to be derived from the NanoLuc domains.

### 3.3. Bioluminescence

Different concentrations of I91_2_-NanoLuc_3_-I91_2_ were used to record bioluminescence. For the bioluminescence measurements we purchased Nano-Glo Luciferase Assay, including NanoLuc’s substrate furimazine, from the Promega Company (Madison, WI, USA) and followed manufacturer instructions. In detail, we freshly prepared assay reagent by mixing 50 volumes of Nano-Glo Luciferase Assay Buffer with one volume of Nano-Glo Luciferase Assay Substrate. This was later used to measure bioluminescence using GlowMax 20/20 Luminometer (Promega Company, E5311).

For the measurements we used 50 μL of various concentrations of I91_2_-NanoLuc_3_-I91_2_, ranging from a few fM to nM, along with 50 μL of the Nano-Glo assay reagent. We waited 3 min prior to collecting the bioluminescence signal to allow the chemistry to equilibrate. Additionally, for the time dependence of the bioluminescence signal, the same volumes of protein and Nano-Glo assay reagents were used. When the two parts of the reaction were mixed, we collected the signal at different time points for the same sample.

## 4. Conclusions

Although NanoLuc was engineered to be thermally more stable and more robust as compared to Firefly Luciferase, our results show it is a mechanically moderately weak protein that is unable to robustly refold itself correctly when denatured. Our results show each NanoLuc repeat in our polyprotein behaving as a single domain protein, with a single mechanical unfolding event occurring on average when a force of 72 pN is applied to the protein. Additionally, we performed cyclic measurements, where the forces applied to a single protein were cyclically increased and lowered to allow the protein the opportunity to refold, and observed the protein was able to refold to its correct structure after mechanical denaturation only 16.9% of the time, while another 26.9% of the time there was evidence of protein misfolding to a potentially non-functional conformation. Because it is so highly luminescent when in its proper conformation, NanoLuc’s bright catalytic activity could potentially be observed during mechanically-induced denaturation and renaturation. As such, it can be potentially useful as a substrate for studies of more complex, multi-component mechanisms of protein refolding [67], such as single-molecule structure/function studies of protein refolding that requires assistance from protein chaperones while simultaneously measuring catalytic activity.

## Figures and Tables

**Figure 1 ijms-22-00055-f001:**
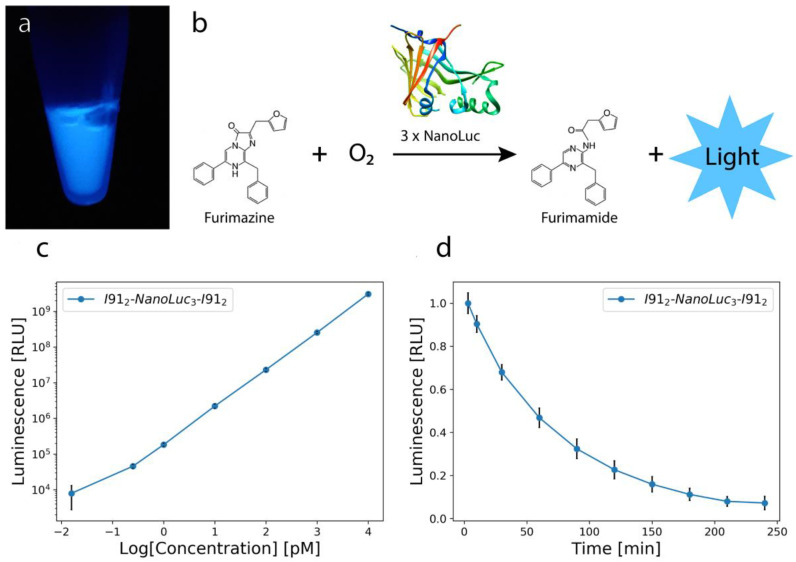
(**a**) Bioluminescence of 50 μL 206 nM I91_2_-NanoLuc_3_-I91_2_ after mixing with 50 μL Nano-Glo Luciferase Assay Buffer. The Assay Buffer contains NanoLuc’s substrate, furimazine. (Imaged by mobile phone) (**b**) ATP-independent bioluminescence reaction catalyzed by NanoLuc during which the substrate (furimazine) is oxidized into furimamide (inspired by Promega company sketch) [86]. (**c**) Bioluminescence results of I91_2_-NanoLuc_3_-I91_2_ construct showing dilutions of 10 nM to ~16 fM vs. logarithmic concentration of the protein. (*n* = 3). (**d**) Normalized time dependence of bioluminescence vs. time of 10 nM I91_2_-NanoLuc_3_-I91_2_ construct (*n* = 4). The mean values were plotted along with the standard error of the mean for the normalized values for both (**c**,**d**) (black vertical lines). All measurements were at room temperature.

**Figure 2 ijms-22-00055-f002:**
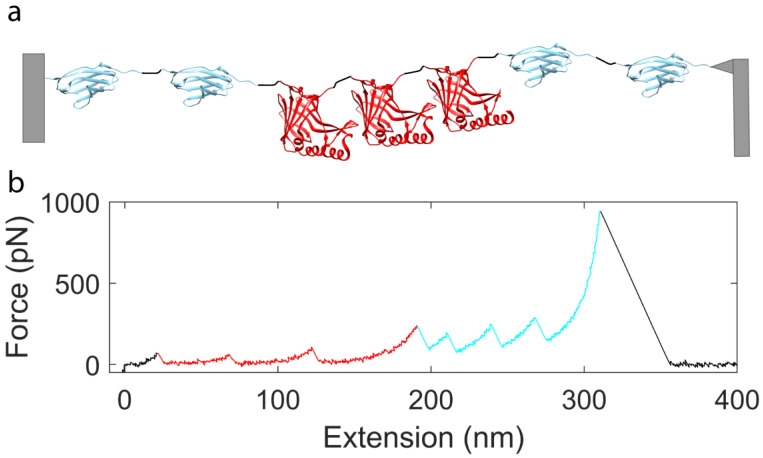
(**a**) Depiction of the I91_2_-NanoLuc_3_-I91_2_ construct. On the far left is the surface and on the far right is the cantilever. In between, we can see with cyan the I91 domain (PDB file 1WAA) and with red the NanoLuc proteins (PDB file 5IBO). The different proteins were connected with short linkers (4–5 residues). (**b**) A full trace of the I91_2_-NanoLuc_3_-I91_2_ construct. Different parts of the unfolding curves are color matched with the respective proteins.

**Figure 3 ijms-22-00055-f003:**
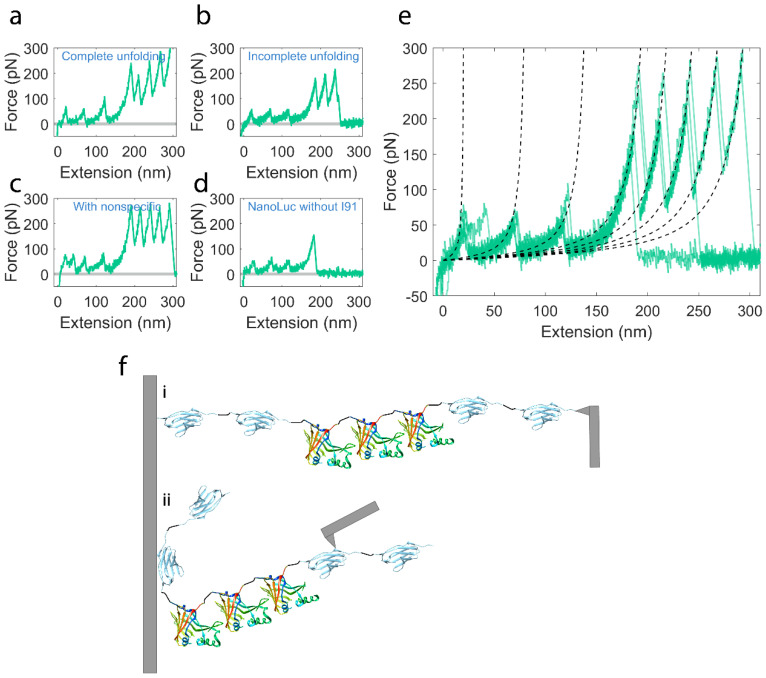
Results of unfolding experiments on I91_2_-NanoLuc_3_-I91_2_ construct. (**a**–**d**) Four representative examples of force–extension curves. (**a**) Complete unfolding curve of I91_2_-NanoLuc_3_-I91_2_ construct. (**b**) Unfolding curve with three NanoLuc events and only three I91 events which is incomplete. (**c**) Unfolding curve with nonspecific events at the beginning. (**d**) Unfolding curve with only unfolding events of NanoLuc. Please note that representative recordings in (**a**–**c**) categories were used in our analysis with *n* = 54. Additionally, even though the number of force peaks attributed to mechanical unfolding of I91, handle domains vary depending on the fragment picked up for stretching (as shown in Figure 3f), the number of small unfolding force peaks that we attribute to NanoLuc repeats is constant. (**e**) Overlay of 4 traces from (**a**–**d**). Dashed lines are worm-like chain (WLC) fits with persistence lengths *p* = 0.4 nm for NanoLuc domains and *p* = 0.3 for I91 domains. (**f**) Illustration of pulling experiment for full unfolding trace in 3a (**i**) and partial unfolding trace in 3d (**ii**). On the far left is the surface and on the far right is the cantilever.

**Figure 4 ijms-22-00055-f004:**
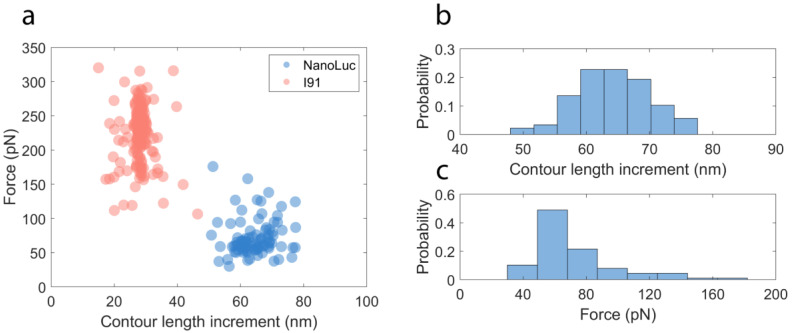
Results of unfolding experiments on I91_2_-NanoLuc_3_-I91_2_ construct. (**a**) Distribution of force and contour length increment of both NanoLuc peaks and I91 peaks. (**b**,**c**) Histograms of the contour length increment (**b**) and the force (**c**) for NanoLuc peaks. Sturge’s formula was used to calculate the number of bins for (**b**,**c**).

**Figure 5 ijms-22-00055-f005:**
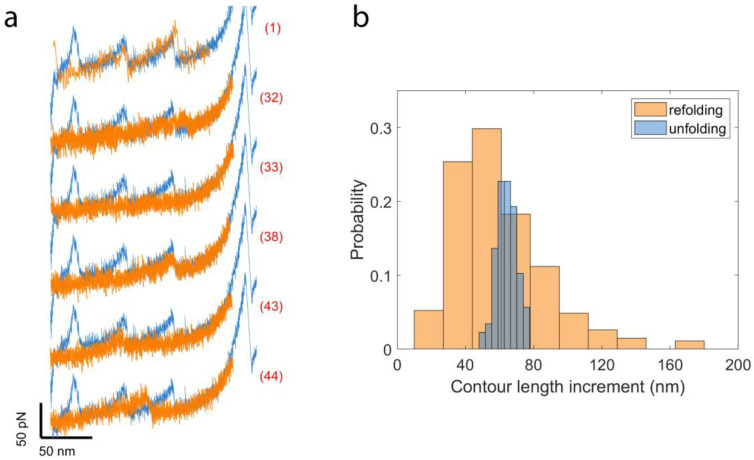
(**a**) Six examples of refolding attempts on the same molecule of I91_2_-NanoLuc_3_-I91_2_ constructed from cycles 1, 32, 33, 38, 43 and 44 (orange line). The complete unfolding curve (blue line) is also plotted for comparison. (**b**) The distribution of the contour length increment from unfolding and refolding experiments. Sturge’s formula was used to calculate the number of bins for (**b**). The histogram of unfolding experiments (Figure 4b) is added for comparison.

**Figure 6 ijms-22-00055-f006:**
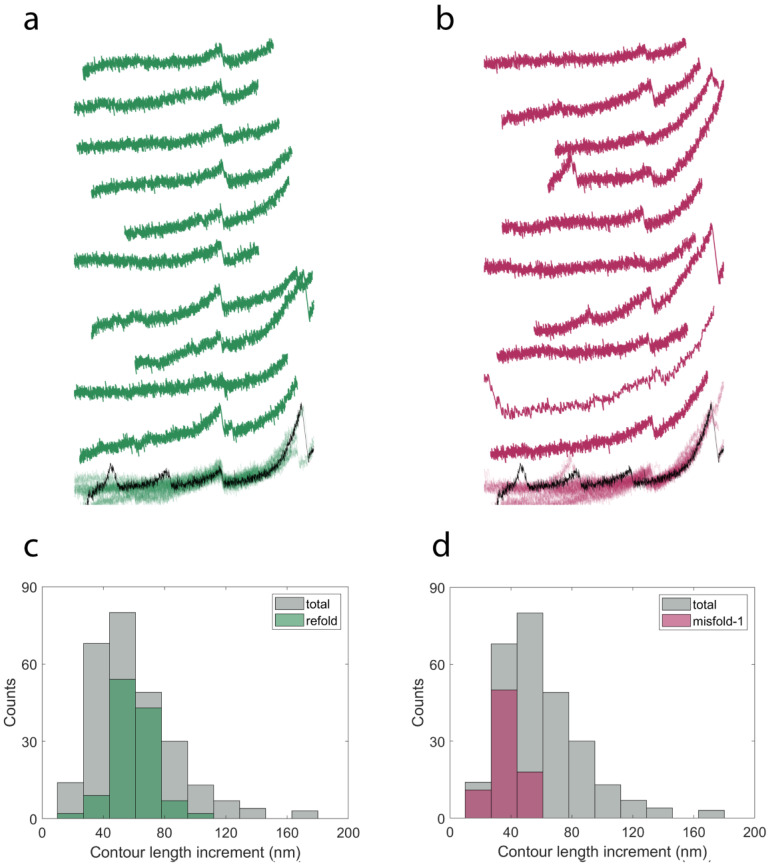
(**a**) Ten examples of curves of refolded proteins. (**b**) Ten examples of peaks with a smaller contour length increments from misfolded proteins. The template is plotted in a black line for (**a**,**b**). The corresponding distributions of contour length increments are shown in (**c**,**d**), respectively. Sturge’s formula was used to calculate the number of bins for (**c**,**d**). Bin size of the total recordings was used for the refold (**c**) and misfold-1 (**d**) results for comparison.

**Figure 7 ijms-22-00055-f007:**
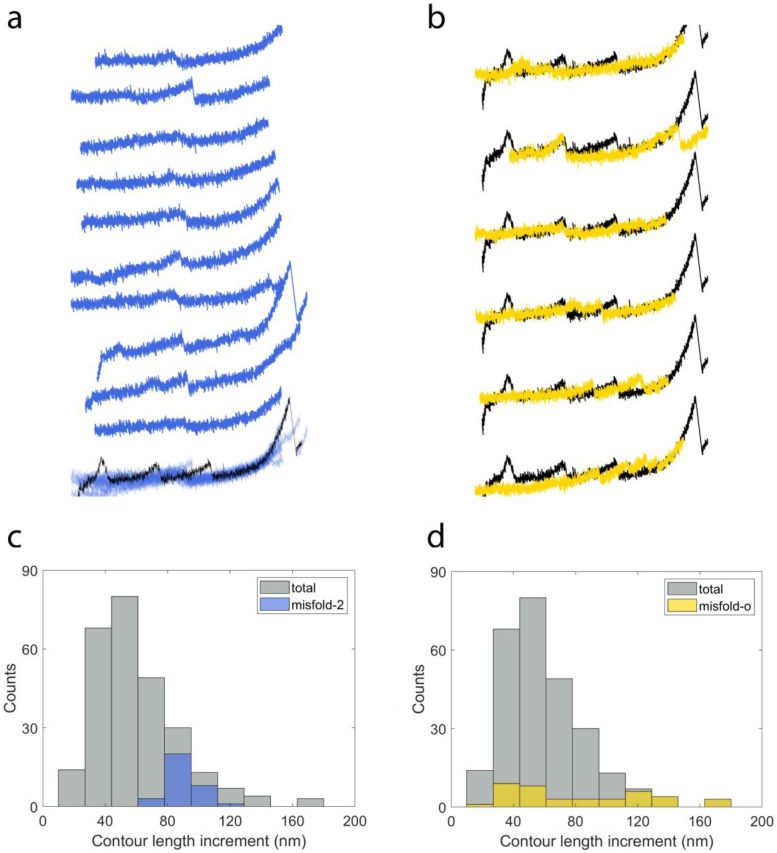
(**a**) Ten examples of curves with peaks of a larger contour length increment. (**b**) Six curves of refolding attempts with different misfolding behavior. The template is plotted in black line. The corresponding distributions of contour length increment are shown in (**c**,**d**), respectively. Sturge’s formula was used to calculate the number of bins for (**c**,**d**). Bin size of the total recordings was used for the misfold-2 (**c**) and misfold-o (**d**) results for comparison.

**Figure 8 ijms-22-00055-f008:**
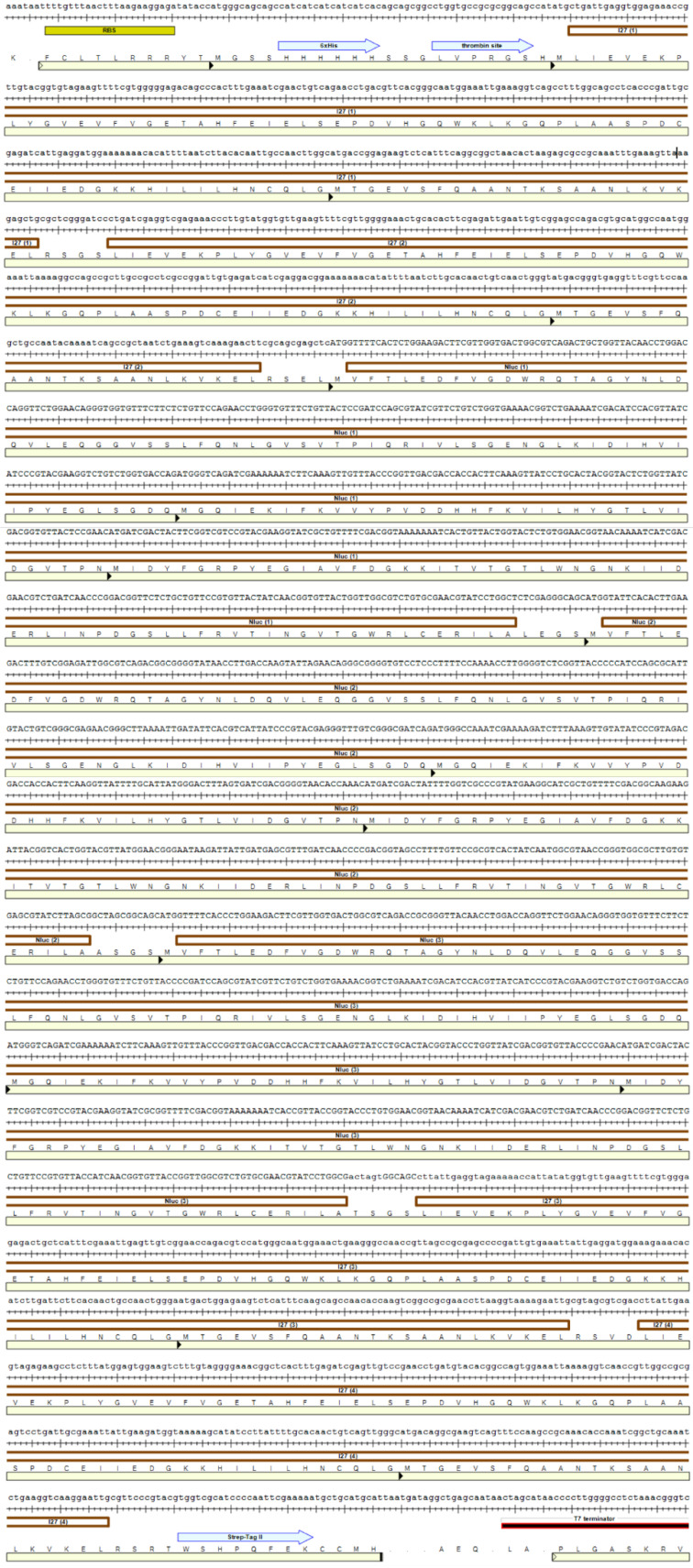
Nucleic acid (5′ to 3′) and amino acid (N to C-terminus) sequence of I91_2_-NanoLuc_3_-I91_2_ construct. Different repeats are marked with a parenthesis with the number of repeat, such as I91 (1). HisTag (6xHis) and StrepTag (Strep-Tag II) are marked with a blue arrow. Proteins are indicated by a brown bar and a light yellow bar is used for the amino acid sequence. DNASTAR program was used.

**Figure 9 ijms-22-00055-f009:**
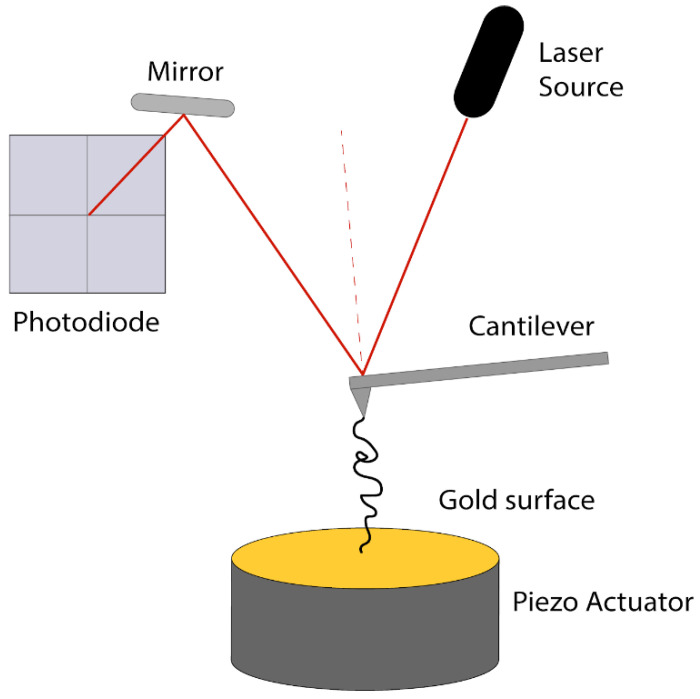
Atomic force microscope (AFM)-based Single Molecule Force Spectroscopy (SMFS) set-up. A laser beam is reflected at the back of the cantilever and then after redirected from a mirror, it is reaching the photodiode. Movement of the cantilever due to pulling of the biomolecule is causing changes to the signal detected by the photodiode. The sample is placed on a gold surface, placed on a piezo actuator, which controls the x-y-z movement of the sample.

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
