# Peer review of "Mechanical Stability of a Small, Highly-Luminescent Engineered Protein NanoLuc"

_ijms, 2020, doi:10.3390/ijms22010055_

Round 1

Reviewer 1 Report

Your manuscript concerning a study on mechanical stability of the luminescent protein NanoLuc describes in a very clear and easy to follow way a well-planned research work, based on single molecule force spectroscopy, on the mechanical stability of a protein. The subject of your study is a protein with luminescent properties and the interest roused by your paper is mainly connected to its application in molecular imaging. To know the stability of such a protein can offer useful information about its performance in different biochemical scenarios. All data you supplied look useful, I just suggest you to remove the Figure 6 since it does not offer further information with respect of what you already reported in the text. 

Author Response

Point 1: Your manuscript concerning a study on mechanical stability of the luminescent protein NanoLuc describes in a very clear and easy to follow way a well-planned research work, based on single molecule force spectroscopy, on the mechanical stability of a protein. The subject of your study is a protein with luminescent properties and the interest roused by your paper is mainly connected to its application in molecular imaging. To know the stability of such a protein can offer useful information about its performance in different biochemical scenarios. All data you supplied look useful, I just suggest you to remove the Figure 6 since it does not offer further information with respect of what you already reported in the text. 

Response 1: We would like to thank the reviewer for his/her positive evaluation of our manuscript. In the revised manuscript we removed Figure 6 as suggested. Additionally, all other figures, following Figure 6, were correctly numbered following the deletion. We also, removed parts in the text where Figure 6 was mentioned. Changes can be found in Lines 263 (deletion of “are included in the pie chart” and replaced by “can be”), Line 270 (deletion of Figure 6), Line 271 (deletion of Figure 6 caption), Line 275 (deletion of “(green color in Figure 6)”), Line 286 (deletion of “(red color in Figure 6)”), Line 293 (deletion of “(blue color in Figure 6)”), Line 312 (deletion of “(yellow color in Figure 6)”). Changes in renumbering the rest of the figures in the text can be found in Line 272 (7a to 6a), Line 276 (7c to 6c), Line 280 (Figure 7 to Figure 6), Line 285 (7b to 6b), Line 290 (7d to 6d), Line 292 (8a to 7a), Line 295 (8c to 7c), Line 300 (Figure 8 to Figure 7), Line 305 (8b to 7b), Line 311 (8b to 7b), Line 314 (8d to 7d).

Reviewer 2 Report

In the manuscript 'Mechanical stability of a small, highly-luminescent engineered protein NanoLuc' by Ding, Apostolidou and Marszalek, the authors report on single-molecule force measurements on the bioluminescent NanoLuc protein. The goal of the study is to investigate the folding/unfolding stability of the relatively recently engineered protein. Performing both unidirectional and cyclic force measurements, the authors are able to determine that the NanoLuc unfolds in a single unfolding event, at a force of about 72 pN, and that only a rather small fraction of about 17% of the unfolded proteins is capable of correct re-folding. This enables them to characterize NanoLuc as mechanically rather weak protein. The paper is a well-written piece of solid science. Considering the large general interest in protein folding in general and in bioluminescent proteins in particular, and their increasingly broad application in biochemistry and biophysics, I am convinced that the paper will find its readership. Provided the authors answer satisfactorily the comments below, I am thus happy to recommend the paper for publication in the International Journal of Molecular Sciences.

1) The distribution of the unfolding forces in Fig. 4C is very interesting. It seems to me that the authors do not do its shape justice fitting it with a Gaussian distribution. Is there some information in the shape of the distribution? What could be the reasons for a (non)-normal distribution?

2) The rainbow coloring of the trace in Fig. 2b with the matching segments in Fig. 2a is very suggestive. Is it really possible to identify the individual parts of the extension curve with the protein secondary structure? If that is so, it seems to provide wealth of detailed information on the protein unfolding, which, in my opinion, deserves to be discussed in the manuscript.

3) Fitting the data by the worm-like chain model enables the authors to extract the contour length increment. As such, it is crucial for the work, and should be described in more detail, including the calculation of the persistence length, either in the supporting information or in the Methods section.  

4) I enjoyed the plot twist, proposing to use the low re-folding probability for NanoLuc as a bright reporter in protein re-folding studies. Could the authors expand further on the expected correlation of the correct re-folding and catalytical activity, possibly citing some literature?

5) Based on the measured unfolding force of 72 pN, the authors categorize the protein as mechanically weak. Could the authors provide brief comparison to similar proteins, for the readers not familiar with the subject?

6) Ref. 1 contains a formatting error, it should read Dobson, C.M., Protein folding and misfolding, Nature 2003, 426, 884-890

7) Page 4 line 129, ‘We later used in our analysis for …’ – some subject is missing in this sentence.

Author Response

Point 1: The distribution of the unfolding forces in Fig. 4C is very interesting. It seems to me that the authors do not do its shape justice fitting it with a Gaussian distribution. Is there some information in the shape of the distribution? What could be the reasons for a (non)-normal distribution?

Response 1: We appreciate the reviewer’s detailed analysis of our results, and specifically Figure 4c. It, indeed, looks as if the distribution is not normal and possibly has peaks at higher forces. However, we note that Single Molecule Force Spectroscopy (SMFS) measurements are of very low success rate and we would like to further increase the number of points in this distribution to be absolutely sure that this distribution is not normal before we start speculating on the origin of the putative extra peaks. We reflected this observation in the manuscript in Line 211 “We note that the distribution in Figure 4c displays non Gaussian features, so further measurements are warranted to clarify their origin.”

Point 2: The rainbow coloring of the trace in Fig. 2b with the matching segments in Fig. 2a is very suggestive. Is it really possible to identify the individual parts of the extension curve with the protein secondary structure? If that is so, it seems to provide wealth of detailed information on the protein unfolding, which, in my opinion, deserves to be discussed in the manuscript.

Response 2: We again appreciate this very important observation. We used the rainbow feature merely to mark NanoLuc proteins and show their internal structure in their N to C direction. We did not mean to imply that there is a direct correlation between rainbow colors in the force peaks and the secondary structure of NanoLuc. Now we understand that this approach is confusing, and readers may try to make this connection. For that reason, we modified Figure 2 (Line 156) to use one solid color for NanoLuc that is different from color used to mark I91. In more general sense, we would like to mention that in some cases it is indeed possible to identify features in the force-extension data with secondary structures and subdomains within a protein, as demonstrated in our previous work on Luciferase, please view Reference 67. However, we do not have that deeper insight for NanoLuc yet and for that reason it would be premature for us to try to make this connection in the main text. We thank reviewer for helping us to improve the clarity of our paper.

Point 3: Fitting the data by the worm-like chain model enables the authors to extract the contour length increment. As such, it is crucial for the work, and should be described in more detail, including the calculation of the persistence length, either in the supporting information or in the Methods section.  

Response 3: We followed reviewer’s recommendation and expanded the section on fitting the experimental data with the worm-like chain model in the revised Methods Line 389. We also added the formula of the WLC proposed by Bustamante et. al and the reference.

Point 4: I enjoyed the plot twist, proposing to use the low re-folding probability for NanoLuc as a bright reporter in protein re-folding studies. Could the authors expand further on the expected correlation of the correct re-folding and catalytical activity, possibly citing some literature?

Response 4: We followed the reviewer’s suggestion and expanded this idea in the introduction. We added on Line 67: “We expect that unraveling of tertiary structure of NanoLuc due to mechanical unfolding should be accompanied by a significant decrease in its bioluminescence activity. Conversely, we expect that successful NanoLuc refolding should restore its ability to produce bright light similar to what is observed in bulk experiments for Firefly Luciferase [81,82]. Thus, in…” As can be seen two references were added.

Point 5: Based on the measured unfolding force of 72 pN, the authors categorize the protein as mechanically weak. Could the authors provide brief comparison to similar proteins, for the readers not familiar with the subject?

Response 5: We agree with the reviewer that 72 pN cannot be considered small or large without providing a reference of what is considered a mechanically strong or weak protein. Actually, this prompted us to make our language more precise and categorize NanoLuc as moderately mechanically weak protein. For comparison, ankyrin repeat proteins unfold at around 20-25 pN [89] and spectrin domains unfold at around 25-35 pN at comparable stretching speeds (~0.3 μm/s) [90]. Both are examples of mechanically weak proteins. On the other hand, some proteins, such as titin domains from muscle, unfold at around 200 pN at the same stretching speeds and those were considered as mechanically strong proteins until the Gaub group recently identified bacterial adhesion proteins that unfold at amazingly high forces of 2,000 pN [91]. We added moderately weak on Line 26 and Line 418 and added the above text on Line 205.

Point 6: Ref. 1 contains a formatting error, it should read Dobson, C.M., Protein folding and misfolding, Nature 2003, 426, 884-890

Response 6: We would like to thank the reviewer for identifying this error. The citation now reads “Dobson, C.M. Protein folding and misfolding. Nature 2003, 426, 884–890.” in Line 444. The change was done through the reference manager program and was not reflected on the “Track changes” like other changes were.

Point 7: Page 4 line 129, ‘We later used in our analysis for …’ – some subject is missing in this sentence.

Response 7: We would like to thank the reviewer for identifying the missing subject in the sentence. In Line 154 we added the subject as follows “We later used persistence length of 0.4 nm in our analysis for NanoLuc force – extension curves.”

Reviewer 3 Report

The manuscript by Ding et al. focuses on force spectroscopy measurements of a chimeric protein with three identical domains (NanoLuc) with luciferase activity. The unfolding and refolding events were followed at a single-molecule level. The data indicate that the NanoLuc is not mechanically stable, despite its reported high thermal stability. From an experimental perspective, the work is a combination of uncorrelated SMFS and biocatalytic measurements. The SMFS results, showing the limited refolding capacity of NanoLuc, merit the publication.

The Referee would like to share the following comments on the manuscript:

  1. It is unclear whether the folding to a fully enzymatically active state occurs for all three NanoLuc domains within recombinant I91(2)-NanoLuc(3)-I91(2) protein. For instance, does the chimeric protein exhibit a three-fold increase in activity in comparison with NanoLuc? Please, comment.
  2. The Referee wonders whether a single-repeat model was tested I91(2)-NanoLuc(1)-I91(2) at all. The reasoning on LL72 is not that clear, “and in order to increase the efficiency of force spectroscopy experiments to study the mechanical properties of NanoLuc, we generated a “polyprotein” gene for three tandem repeats of NanoLuc and flanked them”. What are those limits on the efficiency of experiments? Is it related to “3+4 peaks” signature for the “propper” unfolding track? 
  3. The Referee invites the authors to include appropriate comments in the figure legends concerning “representative” curves, i.e. number of experiments and the similarity of traces between them. 
  4. The bin size on the histograms (especially, Figure 4b) is far from optimal. The histograms in their current form add very little insight into the shape of the distribution plotted in Figure 4a.
  5. Similarly, the Referee is troubled with the justification for fitting these and other skewed results (Figure 7) with the normal distribution. Please provide such an explanation.
  6. Codon usage alterations (L273) are known to influence co-translation folding dynamics and the fraction of correctly folded protein. Please provide complete nucleotide coding sequences for the I91(2)-NanoLuc(3)-I91(2) 
  7. The Referee urges the authors to exclude or amend the last speculative sentence (L27) about the platform's potential utility from the Abstract, since no attempt for simultaneous or correlative SMFS/biocatalysis measurements were made in the work, let alone the connection with chaperones. 

Author Response

Point 1: It is unclear whether the folding to a fully enzymatically active state occurs for all three NanoLuc domains within recombinant I91(2)-NanoLuc(3)-I91(2) protein. For instance, does the chimeric protein exhibit a three-fold increase in activity in comparison with NanoLuc? Please, comment.

Response 1: We would like to thank the reviewer for pointing this out. This prompted us to fix the paragraph starting on Line 77 making it more coherent and clear for the readers. We would like to comment that it is evident that all three NanoLuc proteins fold into mechanically stable and comparably strong structures as observed from force-extension traces. However, at present, we do not know whether all three NanoLuc proteins exhibit full enzymatic activity in the chimeric protein. Clearly, the chimera produces very bright bioluminescence, but the light intensity as measured in our bioluminometer is not 3 x greater as compared to I911-Nanoluc1-I912 construct. Actually, preliminary measurements suggest that I912-NanoLuc3-I912 activity may be lower than I911-Nanoluc1-I912. However, the origin of this effect is presently unknown and needs to be examined in future work, for example by generating NanoLuc1 and NanoLuc3 constructs without flanking I91 domains. We added this comment to the revised manuscript on Line 102, as requested by the reviewer.

Point 2: The Referee wonders whether a single-repeat model was tested I91(2)-NanoLuc(1)-I91(2) at all. The reasoning on LL72 is not that clear, “and in order to increase the efficiency of force spectroscopy experiments to study the mechanical properties of NanoLuc, we generated a “polyprotein” gene for three tandem repeats of NanoLuc and flanked them”. What are those limits on the efficiency of experiments? Is it related to “3+4 peaks” signature for the “propper” unfolding track? 

Response 2: We would like to thank the reviewer for this question. Yes, I911-Nanoluc1-I912 was engineered and tested in preliminary experiments (please see above). The single small unfolding peak that we assign to I911-Nanoluc1-I912 is similar to the unfolding peaks of the construct with I912-NanoLuc3-I912. We revised the text about “the efficiency of experiments” to better reflect what we meant (starting on Line 77). The reviewer is absolutely correct; the polyprotein construct is designed to provide a more robust fingerprint of the protein, which is very useful for new untested proteins, whose mechanical stability is unknown, and a single small unfolding peak can be masked by non-specific adhesive interactions at the beginning of the force-extension recordings or confused with the instrumental drift. The new text explaining this motivation is in Line 86.

Point 3: The Referee invites the authors to include appropriate comments in the figure legends concerning “representative” curves, i.e. number of experiments and the similarity of traces between them. 

Response 3: We would like to thank the reviewer for pointing out our lack of clarity. In response, we modified in Line 182 the caption of Figure 3 as: “Figure 3. Results of unfolding experiments on I912-NanoLuc3-I912 construct. (a-d) Four representative examples of force-extension curves. (a) Complete unfolding curve of I912-NanoLuc3-I912 construct. (b) Unfolding curve with three NanoLuc events and only three I91 events which is incomplete. (c) Unfolding curve with nonspecific events at the beginning. (d) Unfolding curve with only unfolding events of NanoLuc. Please note that representative recordings in (a), (b), and (c) category were used in our analysis with n=54. Additionally, even though the number of force peaks attributed to mechanical unfolding of I91 handle domains varies depending on the fragment picked up for stretching (as shown in Figure 3f), the number of small unfolding force peaks that we attribute to NanoLuc repeats is constant. (e)…”

Point 4: The bin size on the histograms (especially, Figure 4b) is far from optimal. The histograms in their current form add very little insight into the shape of the distribution plotted in Figure 4a.

Response 4: We appreciate the reviewer’s attention to detail. In order to be more consistent, we used Sturges’ formula for bin calculation and we introduced a brief description of this approach on Line 202. Figures 4-7 were accordingly altered with the bin size calculated based on Sturges’ formula. Caption of these figures also includes the use of this formula. Please see Lines 220, 223, 250, 254, 279, 283,

299 and 303.

Point 5: Similarly, the Referee is troubled with the justification for fitting these and other skewed results (Figure 7) with the normal distribution. Please provide such an explanation.

Response 5: We appreciate the reviewer’s detailed analysis of our results, and specifically Figure 4c. It indeed looks as if the distribution is not normal and possibly has peaks at higher forces. However, we note that Single Molecule Force Spectroscopy (SMFS) measurements are of very low success rate and we would like to further increase the number of points in this distribution to be absolutely sure that this distribution is not normal before we start speculating on the origin of the putative extra peaks. We reflected this observation in the manuscript in Line 211 “We note that the distribution in Figure 4c displays non Gaussian features, so further measurements are warranted to clarify their origin.”

Point 6: Codon usage alterations (L273) are known to influence co-translation folding dynamics and the fraction of correctly folded protein. Please provide complete nucleotide coding sequences for the I91(2)-NanoLuc(3)-I91(2) 

Response 6: We know that codon shuffling does not significantly affects I91 fold as tested in our paper reporting on the construction of the new poly(I91) protein (Reference 93). However, we do not presently know whether codon shuffling affects NanoLuc fold and stability. This remains to be tested. We provide the complete nucleic acid sequence for the construct used (Figure 8) in the methods section Line 328 with description in the text on Line 323.

Point 7: The Referee urges the authors to exclude or amend the last speculative sentence (L27) about the platform's potential utility from the Abstract, since no attempt for simultaneous or correlative SMFS/biocatalysis measurements were made in the work, let alone the connection with chaperones. 

Response 7: We thank the reviewer for the suggestion. In response, we changed the sentence on Line 26: “These results show that NanoLuc is a mechanically moderately weak protein that is unable to robustly refold itself correctly when stretch-denatured, which makes it an attractive model for future protein folding and misfolding studies.”. Also, we agree with the reviewer and removed the last sentence from the abstract (Line 28).

Reviewer 4 Report

My comments should be taken with a grain of salt. I am a computational chemist interested in bioluminescence - particularly fluorescent proteins and firefly luciferase. 

  • the bioluminescent aspects of this paper are minimal. They are fine and correct. I was hoping there would be more info about the NanoLuc deformations and misfolding.
  • I don't feel qualified to judge the Single Molecule Force Spectroscopy and that constitutes the vast majority of the paper.
  • Rief has 2 papers in which they examine the stretching of a single fluorescent protein. Having a trimer of NanoLucs seems to complicate the analysis. It is a pity the researchers needed a trimer.
  • Figure 1 should indicate that a trimer is being used. Maybe just a 3 before the NanoLuc on the reaction arrow.
  • This article would be so much more powerful if you could show the gain/loss of bioluminescence associated with unfolding/misfolding and folding. I presume you tried this and had some technical difficulties.
  • My review is solely based on my perspectives as a chemist interested in bioluminescence. As such the paper doesn't have much for me. I hope you have a Single Molecule Force Spectroscopy reviewer.
  • I don't feel qualified to make an overall publish/do not publish recommendation and have choose accept after minor revisions because I need to make a recommendation in order to submit this.

Author Response

My comments should be taken with a grain of salt. I am a computational chemist interested in bioluminescence - particularly fluorescent proteins and firefly luciferase. 

the bioluminescent aspects of this paper are minimal. They are fine and correct. I was hoping there would be more info about the NanoLuc deformations and misfolding.

I don't feel qualified to judge the Single Molecule Force Spectroscopy and that constitutes the vast majority of the paper.

Point 1: Rief has 2 papers in which they examine the stretching of a single fluorescent protein. Having a trimer of NanoLucs seems to complicate the analysis. It is a pity the researchers needed a trimer.

Response 1: We would like to thank the reviewer for this comment. For this first study of NanoLuc’s mechanics we engineered a construct consisting of three tandem repeats of NanoLuc because we did not know what is NanoLuc’s mechanical fingerprint. This “polyprotein” approach is fairly standard in force spectroscopy, as repetitive similar peaks, consistent with construct design, give extra confidence in identifying truly single molecule recordings, as observing a single small peak may be difficult in context of some non-specific adhesive interactions that also break in small unfolding forces. We also engineered a construct with a single NanoLuc flanked by I91 domains and its unfolding single peak is similar to the peaks observed for the polyNanoLuc construct in this manuscript. In future studies we will use both constructs. We note that I911-Nanoluc1-I912 is enzymatically active and produces light.

Point 2: Figure 1 should indicate that a trimer is being used. Maybe just a 3 before the NanoLuc on the reaction arrow.

Response 2: We agree with the reviewer and indicated in Figure 1, as suggested, that “3 x NanoLuc” is used in our bioluminescence and mechanical studies. The previous figure was deleted and the one with “3 x NanoLuc” was used on Line 106.

Point 3: This article would be so much more powerful if you could show the gain/loss of bioluminescence associated with unfolding/misfolding and folding. I presume you tried this and had some technical difficulties.

Response 3: We completely agree with the reviewer and we plan to carry out such experiments in the future. We have not tried this challenging approach yet, as it would be very difficult to do simultaneous force and bioluminescence recordings on an AFM instrument. We hope to make such recordings on a Magnetic Tweezers setup, which we are building, and which naturally uses an inverted microscope and microfluidic chamber, which will allow to record both signals simultaneously. 

My review is solely based on my perspectives as a chemist interested in bioluminescence. As such the paper doesn't have much for me. I hope you have a Single Molecule Force Spectroscopy reviewer.

I don't feel qualified to make an overall publish/do not publish recommendation and have choose accept after minor revisions because I need to make a recommendation in order to submit this.